# Preferential ice growth on grooved surface for crisscross-aligned graphene aerogel with large negative Poisson's ratio

Meng Li[1,2], Nifang Zhao [1], Anran Mao[1], Mengning Wang[1], Ziyu Shao[1], Weiwei Gao [3] ✉ & Hao Bai [1,2] ✉

Ice formation on solid surfaces is a ubiquitous process in our daily life, and ice orientation plays a critical role in anti-icing/deicing, organ cryo-preservation, and material fabrication. Although previous studies have shown that surface grooves can regulate the orientation of ice crystals, whether the parallel or perpendicular alignment to the grooves is still under debate. Here, we systematically investigate ice formation and its oriented growth on grooved surfaces through both in situ observation and theoretical simulation, and discover a remarkable size effect of the grooves. With the designability of surface groove patterns, the preferential growth of ice crystals is programmed for the fabrication of a crisscross-aligned graphene aerogel with large negative Poisson's ratio. In addition, the size effect provides guidance for the design and fabrication of solid surfaces where the effective control of ice orientation is highly desired, such as efficient deicing, long time organ cryo-preservation, and ice-templated materials.

Ice formation on cold surfaces is a ubiquitous phenomenon in nature with wide implications for our daily lives[1,2]. In recent years, ice orientation controlled by selective ice nucleation and preferential ice growth has been the focus of research in various aspects, including anti-icing/deicing[3,4], cell preservation[5], and fabrication of multifunctional materials[6,7], which are related to ice adhesion, cell survival, and formation of porous structure, respectively. Many works reported that oriented ice crystals have been successfully achieved through regulating the intrinsic properties of solid surfaces[8], such as surface microstructure[9–13], surface charge[14,15], and surface wettability[16–18]. Specifically, surface grooves have been proven to be capable of inducing both parallel and perpendicular orientation of ice crystals[19–23]. Lo. et al. reported that water vapor nucleate on the V-grooved surface, preferentially grows along the direction perpendicular to the grooves, and aligns parallelly to the grooves at the macroscopic level[19]. Munch. et al. reported that ice crystals tend to grow parallelly or perpendicularly to the grooves during the freeze-casting process depending on the cold surface[23]. Therefore, ice orientation on grooved surfaces is still under debate, which can be attributed to the limited approach and instruments to directly observe the ice orientation in pure water without any probes. Additionally, the effect mechanism of surface grooves on ice orientation through controlling ice nucleation and growth is still unknown. Therefore, systematic research of ice formation on grooved surfaces is highly demanded.

Here, we investigated ice orientation on various grooved surfaces through both in situ observation and theoretical simulation and proposed a size effect of the grooves. A small amount of dye-labeled graphene oxide nanosheets (0.5 wt.%) were dispersed in water as noninvasive probes to visualize the boundaries of ice crystals. It was found that both parallel and perpendicular preferential ice growth can be achieved on grooved surfaces depending on the groove size. With the designability and versatility of groove patterning, the preferential growth of ice crystals can be facilely achieved and applied to fabricate materials with complex porous architectures and therefore unprecedented properties. As a proof of concept, we fabricated an ultralight graphene aerogel with a unique crisscross-aligned architecture and

[1]State Key Laboratory of Chemical Engineering, College of Chemical and Biological Engineering, Zhejiang University, 310058 Hangzhou, China. [2]Institute of Zhejiang University-Quzhou, 324000 Quzhou, China. [3]Department of Polymer Science and Engineering, Zhejiang University, 310058 Hangzhou, China. ✉ e-mail: wwgao@zju.edu.cn; hbai@zju.edu.cn

ultralow negative Poisson's ratio. Moreover, the groove size effect also provides guidance for engineering solid surfaces to control ice formation, which is highly desired in various aspects, such as efficient deicing, high-percent-survival cell preservation, and ice-templated materials with controllable pore structures.

## Results

### In situ observation of the freezing process on various grooved surfaces

The directional freezing processes on grooved surfaces with various widths (W) and distances (D) were observed using a fluorescent microscope (Fig. 1a and Supplementary Movie 1). Grooves with different size parameters were first generated on silicon wafers by photolithography and subsequently transferred onto a polydimethylsiloxane (PDMS) substrate by soft-lithography (Supplementary Fig. 1). In order to avoid the effect of massive particles on ice growth, all suspensions only contained 0.5 wt.% dye-labeled graphene oxide (GO) nanosheets as the probe of ice boundaries for better observation of ice orientation. On the grooved surface (W = 50 μm,

D = 50 μm; Fig. 1b), ice crystals grew in different directions, ultimately forming a multidomain lamellar structure (Fig. 1e). On the grooved surface (W = 25 μm, D = 5 μm; Fig. 1c), ice crystals grew along the direction parallel to the grooves (Fig. 1f). On the grooved surface (W = 10 μm, D = 25 μm; Fig. 1d), ice crystals grew along the direction perpendicular to the grooves (Fig. 1g). This indicates that the size parameters of surface grooves play a crucial role in the growth direction of ice crystals.

### Groove size effect on the alignment of the freeze-casted structure of materials with different thermal conductivities

In order to investigate the effect of surface grooves on ice orientation, the freeze-casting processes on various grooved surfaces were applied here (Fig. 2a). Various suspensions with different thermal conductivities ($k_{suspension}$; Supplementary Fig. 2 and Table 1) were studied here to investigate the effect of the assembly units on ice growth. In this process, ice crystals nucleated at the cold surface and grew along the vertical temperature gradient, with the units expelled into the space between the adjacent ice crystals. After freeze-drying, large-area

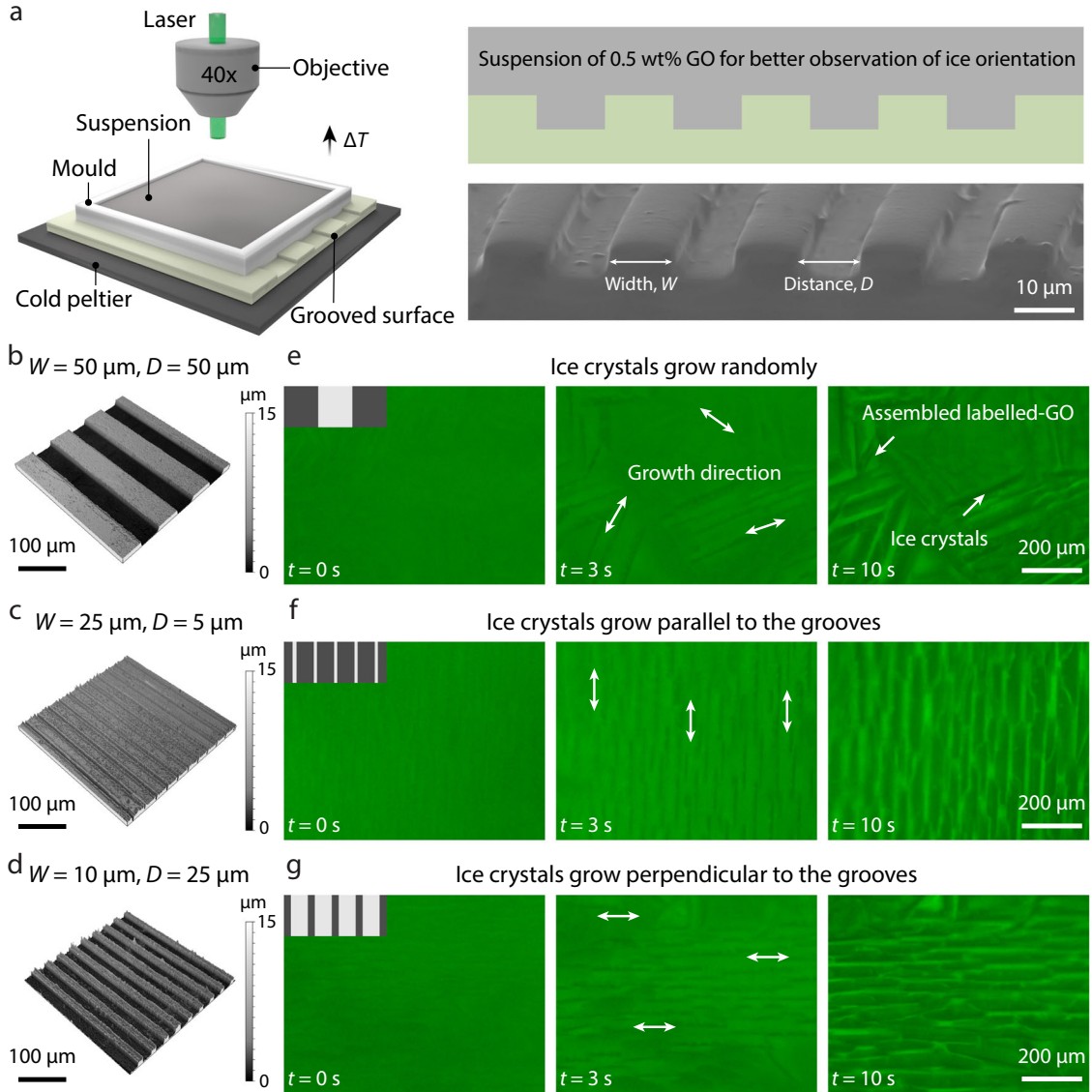

**Fig. 1 | In situ observation of freezing processes on different grooved surfaces.** **a** Scheme of the set-up of in situ observation of ice formation on a grooved surface, two-dimensional scheme of the suspension on the grooved surface, and SEM image of the surface grooves. **b**–**d** Three-dimensional topography of grooved surface with various size parameters. **e**–**g** Successive fluorescent microscopic images showing the ice growth behavior on various grooved surfaces in **b**–**d**. Dark regions represent ice crystals, and bright regions represent aggregated GO nanosheets. The scale bar applies to all images in the panel.

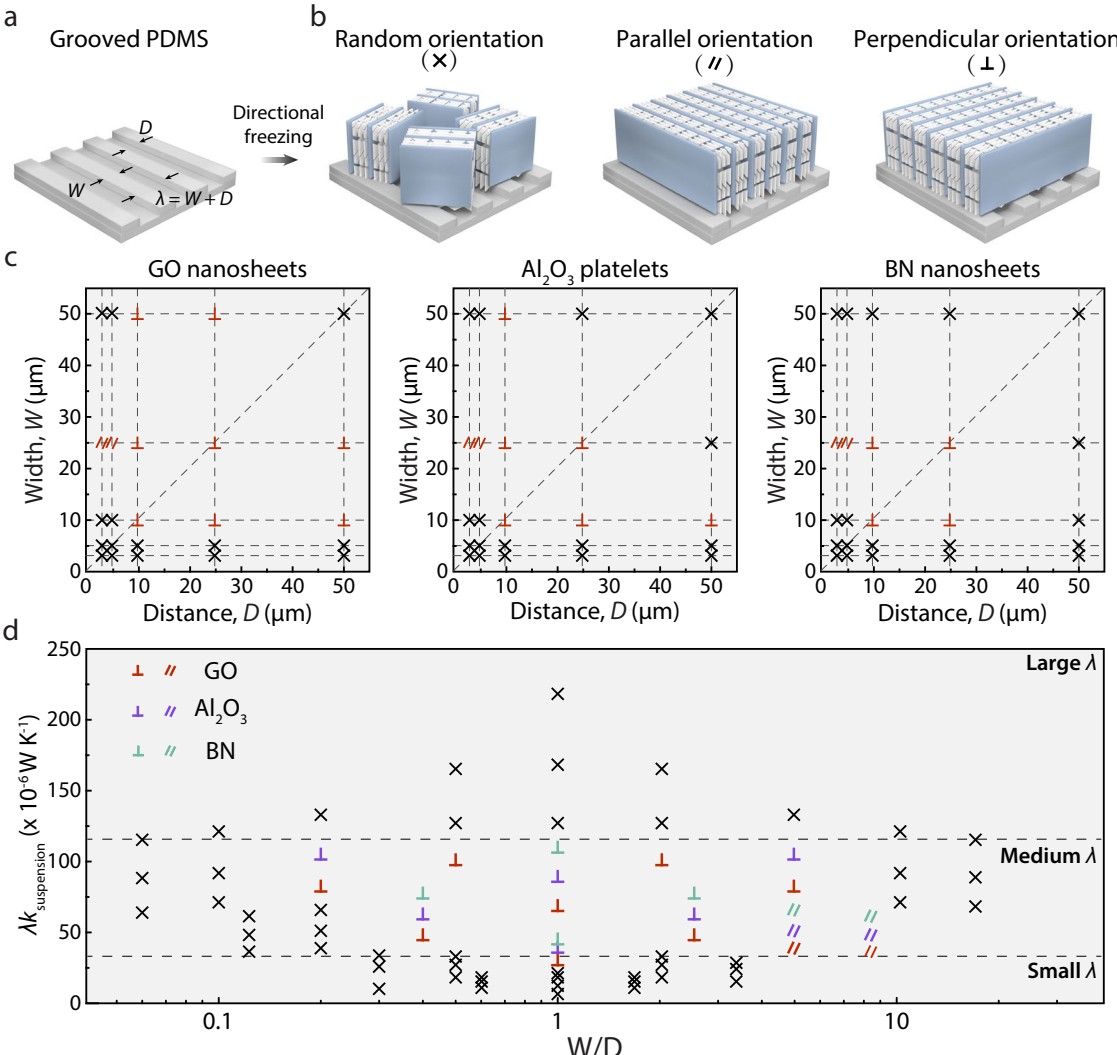

**Fig. 2 | Effects of groove size on the alignment of the freeze-casted structure from materials with different thermal conductivities. a** Scheme of the grooved PDMS, $W$ refers to the width of the convex part of the groove, $D$ refers to the width of the concave part of the groove, and $\lambda$ is the width of one groove composed of one convex part and one concave part. **b** Schemes of three ice orientations, including random orientation, the orientation parallel and perpendicular to surface grooves. **c** Summarized alignment of the microstructures obtained by freezing on grooved PDMS surfaces with various $W$ and $D$. **d** The phase diagram of the $\lambda k_{suspension}$ and $W/D$. The symbol of crisscross lines (×) refers to random orientation, the symbol of parallel lines (∥) refers to the orientation parallel to the grooves, and the symbol of vertical lines (⊥) refers to the orientation perpendicular to the grooves, the definition of the symbols is applied for all colors. Source data are provided as a Source Data file.

lamellar structures of graphene oxide (GO) nanosheets, alumina ($Al_2O_3$) platelets, and boron nitride (BN) nanosheets were obtained and shown by the scanning electron microscopy (SEM) images observed in the cross-section parallel to the grooved surface (Supplementary Fig. 3). These results directly reflect the ice orientation generated on the corresponding grooved surfaces and also demonstrate that the preferential orientation of ice crystals on the grooved surface can be achieved with various suspensions.

A series of substrates with a wide combination of grooves $W$ and $D$ were then systematically studied, and the resulting alignment was analyzed and classified into three categories, including small-area multidomain (×), large-area monodomain parallel (∥), and perpendicular (⊥) to the grooves (Fig. 2b). As shown in Supplementary Fig. 4, SEM images were processed with Image J software to generate gray value-angle curve. When parallel or perpendicular orientation was obtained, the curve exhibits two main peaks, while when random orientation was obtained, the curve exhibits multiple peaks. Figure 2c summarizes the results of the systematic exploration. On one hand, parallel alignment can be only achieved on the grooved surface with

small $D$ and large $W$, in other words, a large $W/D$. On the other hand, perpendicular alignment can be obtained on the grooved surface with $W$ and $D$ in a specific range, and the specific points at the distance-width map are all diagonally symmetrical, which can be correlated with the periodicity ($\lambda = W + D$). As shown in Supplementary Fig. 5, parallel and perpendicular alignments in the $\lambda$-$W/D$ map distribute in a concentrated region, which varies among the suspensions with different $k_{suspension}$. Therefore, we infer that ice orientation is controlled by $\lambda$, $k_{suspension}$, and $W/D$. As shown in Fig. 2d, when $\lambda k_{suspension} < 30$ ($\times 10^{-6}\,W\,K^{-1}$) (small $\lambda$) or $\lambda k_{suspension} > 120$ ($\times 10^{-6}\,W\,K^{-1}$) (large $\lambda$), ice crystals grow into small-area alignment. When $30 < \lambda k_{suspension} < 120$ ($\times 10^{-6}\,W\,K^{-1}$) (medium $\lambda$), the growth mode of ice crystals is dependent on the ratio of $W$ and $D$. This phase diagram indicates that the desired ice orientation in the suspension with a certain thermal conductivity can be obtained under the induce of surface grooves with a certain $W$ and $D$ (calculated by $W/D$ and $\lambda$), which holds great promise for the design and fabrication of functional grooved surface. It should be noted that with higher thermal conductivity, smaller $\lambda$ is required to effectively generate preferential orientation. When $\lambda$ is at nanometer

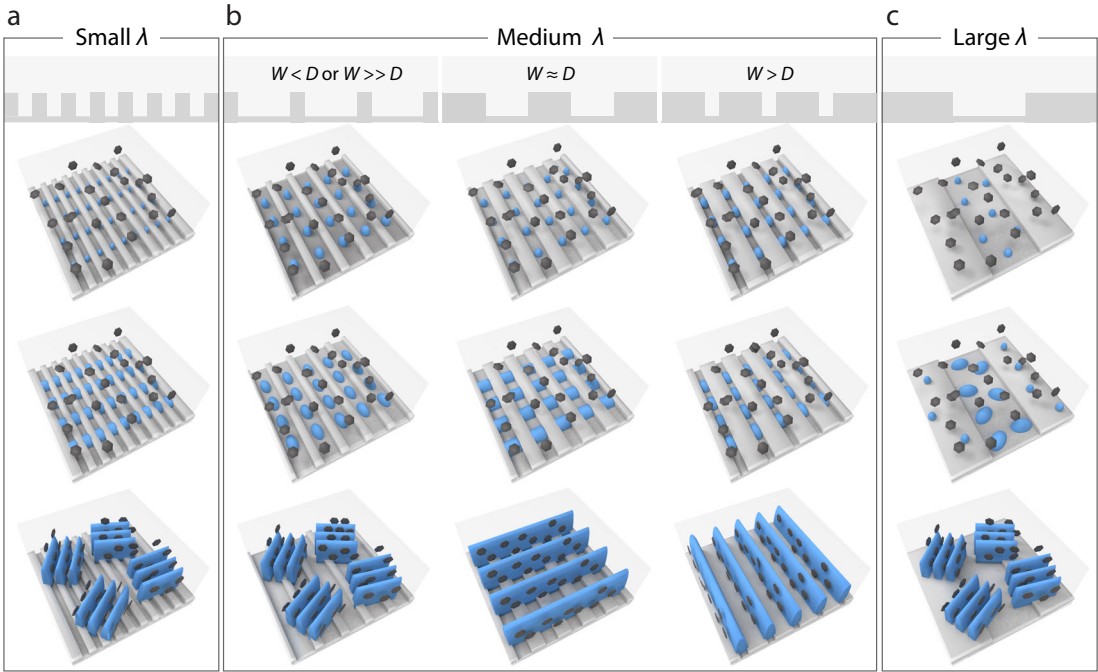

**Fig. 3 | Proposed freezing mechanism. a–c** Schematic illustration of the freezing process on various PDMS substrates. **a** On the grooved surface with small $\lambda$, ice crystals nucleate at the bottom of the grooves and grow randomly, forming a multidomain pattern with small-area alignment. **b** On the grooved surface with medium $\lambda$, the growth modes of ice crystals depend on the $W/D$. **c** On the grooved surface with large $\lambda$, ice crystals nucleate and grow randomly at the bottom of the grooves, forming a multidomain pattern with small-area alignment. The blue units represent the ice crystals, the dark gray symbols represent the dispersed units in the suspensions, the light gray symbols represent the grooved substrate, and the transparent boxes represent the suspensions.

scale, other mechanisms like nucleation barrier difference may play a more important role and new mechanism should be developed.

**Proposed freezing mechanism**

According to the heterogeneous nucleation rate, $J = J_O \, exp(-\Delta G \, k_B^{-1} \, T^{-1})$[8], ice crystals preferentially nucleate at the bottom of the grooves with a lower temperature. Additionally, as our grooves are around several to tens of micrometer scale, three orders of magnitude than the critical radius of ice crystals[24], we consider the grooves size have a negligible effect on nucleation energy barrier. According to the experimental results and theoretical analysis, typical freezing processes on grooved surfaces with different sizes are illustrated in Fig. 3. For the grooved surface with small $\lambda$, ice crystals nucleate at the bottom of the grooves and grow randomly towards different directions, resulting in a multi-domain structure with small-area alignment (Fig. 3a). For the grooved surface with large $\lambda$, ice crystals nucleate and grow randomly at the bottom of the grooves, which is similar to the conventional freezing process. For the grooved surface with medium $\lambda$, the growth modes of ice crystals are dependent on the ratio of $W$ and $D$ (Fig. 3b). When $W \gg D$ or $W < D$, ice crystals nucleate and grow at the bottom of the grooves until they randomly merge together into a small-area alignment. When $W \approx D$, ice crystals nucleate at the bottom of the grooves and grow perpendicularly to the groove, resulting in a monodomain structure with large-area alignment. It is well-known that temperature gradient is the main factor to control ice growth. We hypothesize that such distinct behaviors of preferential ice growth can be attributed to the localized temperature gradient on the grooved surface[7]. When $W > D$, ice crystals nucleate at the bottom of the grooves and grow towards random directions until ice crystals in a single groove merge together, resulting into a large-area lamellar architecture aligned parallelly to the grooves, which is a common growth mode for ice crystals on the grooved surface. It can well account for this kind of preferential growth of ice crystals that the merging behavior would occur more easily among ice crystals in a single groove than the adjacent grooves due to

the geometric constraint by the grooves. Therefore, the geometric constraint and the localized temperature gradient are attributed for the parallel and perpendicular preferential growth of ice crystals, respectively.

In order to explain the perpendicular ice growth, we carried out theoretical simulations to describe the temperature field on various grooved surfaces, as it is challenging to experimentally detect this localized temperature gradient within grooves. Since perpendicular orientation occurs on the grooved surfaces with $W \approx D$, therefore grooved surfaces with various $\lambda$ and $W = D$ are investigated here. Different from the homogeneous temperature distribution on the smooth surface, the localized temperature difference between the concave and convex parts is obvious on the grooved surface and the maximum temperature gradient is generated along the direction perpendicular to the grooves (Supplementary Fig. 6). Therefore, the temperature field at $yz$ plane was investigated to acquire the maximum temperature gradient on various grooved surfaces (Fig. 4a). Because the growth tendency of ice crystals at the horizontal plane of the grooves' top ($z = H$) determines the ultimate orientation, the temperature distribution at this place was quantitatively evaluated. This temperature variation is plotted with respect to the corresponding $y$ position for $\lambda = 20 \, \mu m$ at different $t$. As illustrated in Fig. 4c, $T_1$ and $T_2$ represent the temperature of the midpoint at the top of the concave and convex parts, respectively. Figure 4d shows the dependence of $T_1$ and $T_2$ on $\lambda$ when ice crystals grow up to the top position of the concave parts ($t = t_1$) (Fig. 4b). The corresponding $\nabla T$ is calculated by $(T_2 - T_1)/(\lambda/2)$, representing the localized temperature gradient along the $y$ axis (Fig. 4e). It shows that the perpendicular orientation (red line) can be achieved with grooves in a narrow range of $\lambda$, which can generate the temperature gradient large enough (>1.5 K/cm) to promote the preferential growth of ice crystals perpendicular to the grooves. It is noteworthy that the specific periodicity is closely relevant to the freezing conditions, such as the cooling rate, the thermal conductivity of the cold substrate, and suspension properties. Figure 4e

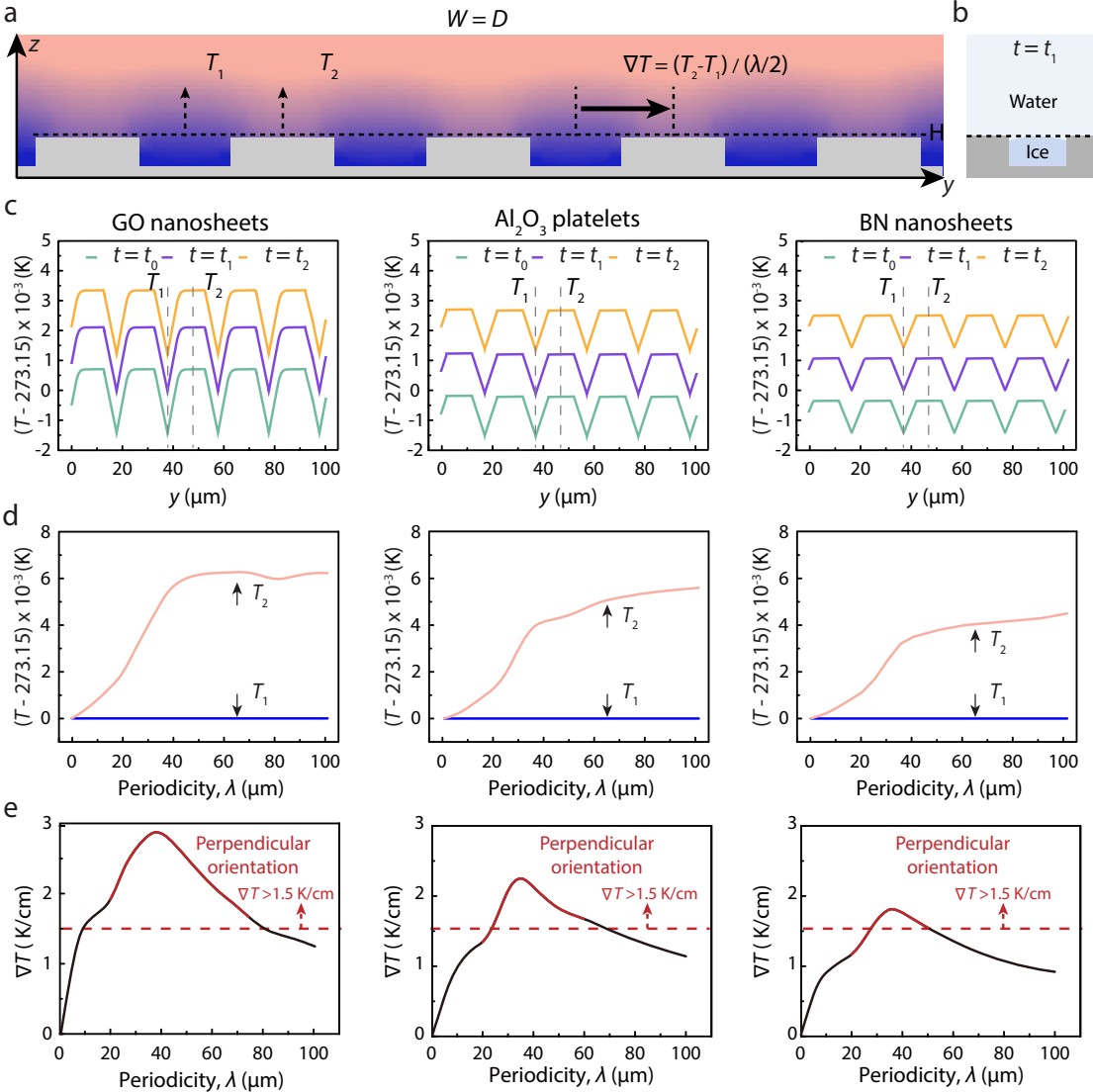

**Fig. 4 | Simulation analysis of temperature field on the grooved surface.**
**a** Scheme of the theoretical simulation showing the typical temperature distribution on the grooved surface. The blue color represents low temperature and the red color represents high temperature. $T_1$ and $T_2$ are the temperatures at the center of the concave and convex parts at $z = H$. **b** Scheme of the freezing process when ice crystals grow up to the top of the concave parts. **c** Temperature distribution along $y$ position at different $t$ at $z = H$ when $\lambda = 20$ μm. **d** Plot of $T_1$ and $T_2$ with respect to $\lambda$ at $t = t_1$. **e** Plot of the localized temperature gradient ($\nabla T = (T_2 - T_1)/(\lambda/2)$) with respect to $\lambda$, red regions show the effective $\lambda$ that can induce the formation of the perpendicular orientation of ice crystals. Source data are provided as a Source Data file.

shows that a wider range of specific periodicity of surface grooves can achieve the preferential growth of ice crystals in the suspension with low thermal conductivity.

Additionally, in order to further prove the formation of three growth modes, the simulation of the temperature field on the grooved surfaces with a constant $\lambda$ (30 μm) but different $W/D$ was performed. As shown in the Supplementary Fig. 7, for the grooves with $W = 25$ μm and $D = 5$ μm, there is no significant temperature gradient to induce the preferential growth of ice crystals, thus ice crystals nucleate at the bottom of the grooves and grow towards random directions until ice crystals in a single groove merge together due to the geometric constraints. For the grooves with $W = 5$ μm and $D = 25$ μm, there is also no significant temperature gradient to induce the preferential growth of ice crystals, thus ice crystals nucleate at the bottom of the grooves and grow until they randomly merge together into a small-area alignment without geometric constraints. For the grooves with $W = D = 15$ μm, an obvious temperature gradient was generated to induce the perpendicular growth of ice crystals. Therefore, we propose that the

preferential growth of ice crystals on the grooved surface can be programmed through either the geometric constraint or a localized temperature gradient.

## Unique porous material fabricated on surface with a special groove pattern

It is well-known that the structure of freeze-casted materials plays the critical role in the final property, thus many researches have been focusing on the tailoring of freeze-casted architecture and developed various techniques, such as external field assisted freezing[25,26], under-flow freezing[27], bidirectional freezing[7], and surface wettability induced freezing[18]. In this work, surface grooves induced freezing was investigated based on the localized temperature gradient between two microgrooves, holding great potentials for the construction of complex porous architectures.

In order to demonstrate the potential application of our discovery, porous material with a unique architecture was fabricated by freeze-casting on a grooved surface with a composite pattern. In

particular, a cross-aligned structure (Supplementary Fig. 8e) was obtained by freeze-casting on the surface with grooves in cross orientation (Fig. S8a, b). The SEM images of the cross-lamellar structure taken from the cross-section perpendicular to the cold surface show that the lamellar layers keep consistent throughout the observed height (Fig. S8c, d). These results indicate the possibility of tailoring the architecture of the freeze-casted materials by harnessing the rich designability of surface grooves by photolithography, generating an effective and scalable approach for the assembly of functional building blocks into bulk materials with complex architectures and multifunctionality.

In recent years, aerogels have been extensively explored due to their excellent thermal[28,29], electrical[30–32], and mechanical performance[33–35] at ultralow density. An extra negative Poisson's ratio (NPR) behavior would endow aerogels with a substantial enhancement of mechanical robustness to resist damage through volume shrinkage when under inevitable deformation,[36–38] which requires precise design for the porous architecture of the aerogel. As shown in Fig. 5a, a reentrant porous architecture, defined as the crisscross alignment, is a typical NPR construction. In order to achieve an ultralow NPR in the aerogel, the effect of the angle ($\theta$) between the lamellae orientation and vertical compression direction on the NPR behavior was firstly investigated through a finite element simulation (Fig. 5b, Supplementary Fig. 9, and Supplementary Movie 2). Figure 5b shows the simulated compression process of the typical architecture with $\theta = 45°$(pattern-45). As depicted in Fig. 5c, the ratio of the NPR minimum of the crisscross architecture with different $\theta$ and $\theta = 45°$ shows that pattern-45 generates the lowest NPR value.

As a proof of concept, graphene aerogel with the typical architecture ($\theta = 45°$) has been successfully produced through the controllable ice growth induced by the surface grooves with corresponding pattern (Fig. 5d, e). This aerogel exhibits a minimum ($-0.55$) of NPR and ultralow density ($3 \text{ mg cm}^{-3}$), outperforming those of previously reported graphene oxide-based (GO-based) aerogels[36–41] (Fig. 5f, g, Supplementary Fig. 10, and Table 2). For comparison, anisotropically radial-aligned graphene aerogels were also fabricated and have an NPR minimum of only $-0.18$, which results from the lamellae orientation with numerous angles (Supplementary Fig. 11). For crisscross-aligned graphene aerogel, it shows a maximum compressive stress of 4.5 kPa when compressed at 99% and still retains 70% of the maximum stress after 100 cycles (Supplementary Fig. 12a, b). However, radial-aligned aerogel only retains 30% of the maximum stress after 100 cycles (Supplementary Fig. 12c, d). This indicates that the lower negative Poisson's ratio endows the crisscross-aligned graphene aerogel with much more excellent mechanical robustness (Supplementary Movie 3). Besides, compared to widely-developed NPR materials with isotropic reentrant architecture, ultralight graphene aerogels with anisotropic architecture holds great potentials in thermal insulation (Supplementary Fig. 13 and Table 3)[29,39,42–44]. As shown in Fig. 5h, our crisscross-aligned graphene aerogels exhibit an ultralow thermal conductivity ($12.1 \text{ mW m}^{-1}\text{ K}^{-1}$) comparable to the commercial silica aerogel, and maintain excellent thermal insulating property after compression and recovery from a 90% strain, while the commercial aerogel is broken after compression. Notably, such programmable ice growth on the grooved surface provides an efficient route for the fabrication of multifunctional graphene aerogels with tunable NPR behaviors, such as soft actuators with positive Poisson's ratio, sensors with zero Poisson's ratio, and body armors with negative Poisson's ratio.

In summary, we experimentally and theoretically investigated the mechanism of preferential ice growth on various grooved surfaces. Parallel and perpendicular preferential ice growth were discovered through in situ observation of water freezing with dye-labeled GO nanosheets as the probe of ice boundaries. The size effects of surface grooves, including the high-probability merging behavior of ice crystals in a single groove and localized temperature gradient, were proposed to account for the parallel and perpendicular alignment, respectively. Our research highlights the potential of utilizing the rich designability of surface grooves to fabricate porous materials with complex architecture and multiple functions through the ice-templating process. As a proof of concept, a unique surface groove pattern was utilized to successfully fabricate GO aerogel with crisscross lamellar architecture and unprecedented negative Poisson's ratio. In addition, this process can also be extended to assemble various building blocks including ceramic nanoparticles, polymers, and functional nanosheets, which are required for many strategic applications including buildings, transportation, and the aerospace industry. More importantly, our discovery of the groove size effect provides powerful guidance for the design and fabrication of solid surfaces to control ice orientation, which holds great promise in the application fields, such as fast and efficient deicing, long-life and high-percent-survival cryo-preserved cell and organs, as well as ice-templated materials with aligned pores.

## Methods

### Materials

The grooved silicon wafer was fabricated by photolithography by commissioning Suzhou Research Materials Microtech Co., Ltd., China. Polydimethylsiloxane (PDMS, Sylgard 184) was purchased from Dow Corning Co., Ltd., USA. Ethanol (AR) and sodium bicarbonate (AR) were purchased from Sinopharm Chemistry Co., Ltd., China. Octadecyltrichlorosilane (OTS) was purchased from Macklin Biochemical Co., Ltd., China. Graphene oxide sheets (diameter: 0.5–5 μm; thickness: 0.8-1.2 nm, 99%) were purchased from Hangzhou Gaoxi Tech Co., Ltd., China. The alumina platelets (density = $3.94 \text{ g cm}^{-3}$, Rona FlairTM White Sapphire) were purchased from Merck, Germany. Boron nitride (BN) powders with average size of 10 μm were purchased from Dandong Chemical Engineering Institute Co., Ltd., China. Poly(vinyl alcohol) (PVA, $M_w$ ~ 205,000), 5-([4,6-dichlorotriazin-2-yl]amino)fluorescein hydrochloride (5-DTAF), and dimethyl sulfoxide (DMSO) were purchased from Aladdin Chemistry Co., Ltd., China. Waterborne polyurethane solution (PU, 35 wt.%) was purchased from ShunDe SanSheng Trade Co., Ltd., China. Poly(ethylene glycol) (PEG-300) was purchased from Sigma-Aldrich Chemistry Co. Ltd., USA. Darvan 811 was purchased from R.T. Vanderbilt Co., Norwalk, CT, USA.

### Fabrication of grooved PDMS substrate

The photo-lithographical silicon wafer was cleaned by ultrasonic washing in a water bath for 10 min and dried with inert gas. The neat silicon wafer was put in a desiccator with a drop of OTS (5 μL) under the vacuum atmosphere and 70 °C for 2 hours. The modified surface turned out to be hydrophobic. Then the silicon wafer was put in an aluminum plate and PDMS was poured to cover the silicon wafer. After curing at 70 °C for 2 hours, the grooved PDMS substrate that duplicated the pattern on the silicon wafer could be obtained. The PDMS substrate is 1 mm in thickness and the grooves are 5 μm in height, respectively.

### In situ observation of the freezing process

In order to observe ice orientation, we added 0.5 wt% graphene oxide (GO) nanosheets into the water, and the GO nanosheets made the ice boundaries clear. At the same time, in order to clearly observe GO nanosheets, we labeled them with fluorescent dyes and observed them using fluorescence microscopy. Firstly, 1 mL GO dispersion ($10 \text{ mg mL}^{-1}$) was added to 100 mL sodium bicarbonate (0.1 M, pH = 9) solution. 5 mg of 5-DTAF dissolved in 1 mL of anhydrous DMSO was further added into 100 mL of the sodium bicarbonate solution to form a reactive dye solution. And the solution was stored for 12 h under 4 °C in a dark environment to form conjugated fluorochromes. Finally, dye-labeled graphene oxide dispersions were obtained by

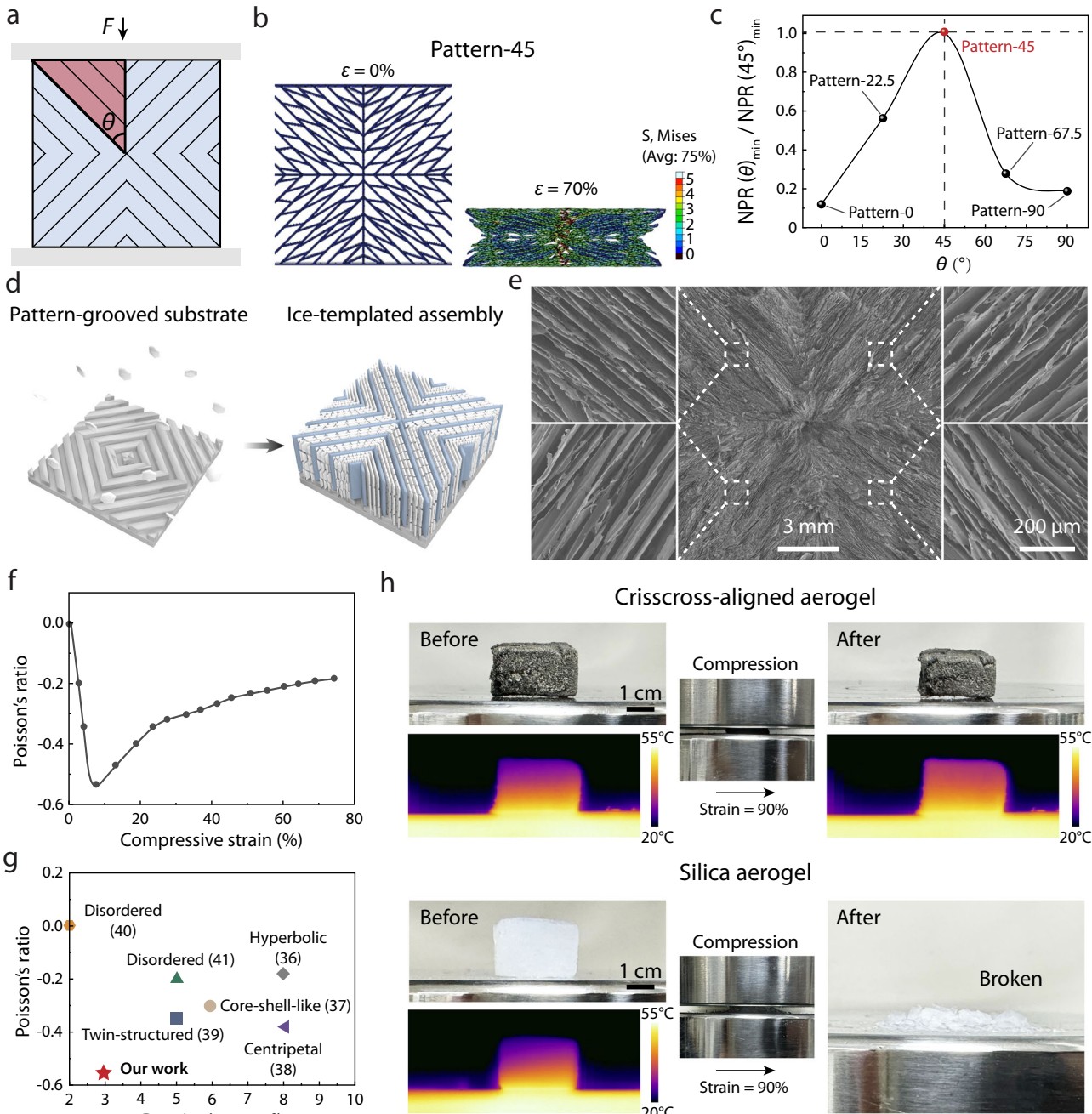

**Fig. 5 | Negative Poisson's ratio behavior of ice-templated graphene aerogels. a** Scheme of the compression process, $\theta$ is the angle between the lamellae orientation and vertical compression direction, *F* is the force applied on the porous materials. **b** Finite element numerical simulation of the compression process of the typical crisscross-patterned aerogels, before being compressed (left) and after being compressed to the compression strain ($\varepsilon$) of 70% (right), different colors represent different stress distribution. **c** Plot of the $\theta$ and NPR$_{min}$/NPR(pattern-45)$_{min}$ obtained through the finite element simulation, NPR$_{min}$ refers to the NPR minimum of every architecture, NPR(pattern-45)$_{min}$ refers to the NPR minimum of the architecture with $\theta = 45°$. **d** Schematic illustration of the pattern-grooved surface-induced freeze-casting process. **e** SEM images of the crisscross-aligned graphene aerogel taken at the cross-section parallel to the surface, the scale bar applies to all SEM images inset. **f** The evolution of the Poisson's ratio as a function of the compressive strain. **g** Ashby map of the Poisson's ratio and density of the crisscross-aligned aerogel and previously reported graphene-based aerogels. **h** Optical and infrared images of the crisscross-aligned graphene aerogel and commercial sol-gel silica aerogel before and after being compressed. Source data are provided as a Source Data file.

centrifuging multiple times with deionized water to remove excess dye molecules.

**Fabrication of 3D porous scaffold**

The GO/PVA suspension was prepared with GO (3 mg mL$^{-1}$) and PVA (3 mg mL$^{-1}$). The suspension was sonicated for 1 hour at 60% power and degassed before freezing. Alumina suspension was prepared by mixing distilled water with alumina platelets (20 vol.%), Darvan 811 (2 wt.% to alumina), poly (ethyl glycol) (2 wt.% to alumina), which were regarded as the dispersant and the lubricant respectively. The suspension was ball-milled for at least 24 hours and degassed before freezing. The BN/PU suspension was prepared by mixing distilled water with exfoliated BN nanosheets (10 vol.%) and PU solutions (25 wt.% to BN). The suspension was sonicated for 1 hour at 60% power and degassed before

freezing. Square Teflon tubes were placed on the grooved PDMS substrate with various groove sizes. The suspension was poured into the mold and frozen on the grooved PDMS substrates at a cooling rate of 5 °C min⁻¹ from −30 °C. After being entirely frozen, the sample was freeze-dried for more than 24 hours with freeze dryer under 0.05 mbar pressure (Labconco 8811, Kansas City, USA). The commercial aerogel is silica aerogel fabricated by sol-gel method, having a thermal conductivity of around 16 mW m⁻¹ K⁻¹.

## Characterization

Scanning electron microscopy (SEM) images were collected by Hitachi S-3500N at an acceleration voltage of 5 kV. The thermal conductivity $k$ was calculated according to the equation $k = \alpha\rho C$, where $\alpha$, $\rho$, and $C$ are the thermal diffusivity, density, and specific heat capacity, respectively. The thermal diffusivity $\alpha$ was measured through the Hot Disk TPS 2500 S in the transient anisotropic mode. All the experiments and simulations were performed under the same condition. The relative humidity was around 60%, the environment temperature was around 25 °C, and the pressure was 1 atm. For each category, at least five replicates were prepared to obtain reliable data.

## Data availability

The authors declare that the data supporting the findings of this study are available within the article and Supplementary Information. Additional datasets related to this study are available from the corresponding author upon request. Source data are provided with this paper.

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

## Acknowledgements

We gratefully acknowledge financial support from the National Natural Science Foundation of China (No. 22075244 by H.B.), Shanxi-Zheda Institute of Advanced Materials and Chemical Engineering (No. 2021SZ-TD009 by H.B.), Zhejiang Provincial Natural Science Foundation of China (No. LZ22E030001 by H.B.), Science and Technology Program of Institute of Zhejiang University-Quzhou (No. IZQ2021KJ2001 by H.B.), and Fundamental Research Funds for the Central Universities (2021FZZX001–17 by W.G.).

## Author contributions

H.B. conceived the concept and supervised the project. M.L., N.Z., and M.W. performed the experiments. Experimental results were analyzed through contributions of M.L., N.Z., A.M., M.W., and Z.S.; H.B., W.G., and M.L. wrote the manuscript.

## Competing interests

The authors declare no competing interests.
