## [Peer Review File · Nature Communications]

Preferential ice growth on grooved surface for crisscross-aligned graphene aerogel with large negative Poisson's ratioEditorial Note: Parts of this Peer Review File have been redacted as indicated to remove third-party material where no permission to publish could be obtained.

REVIEWER COMMENTS

Reviewer #1 (Remarks to the Author):

In this work, the authors investigated the ice growth on grooved surfaces. It was found that parallel and perpendicular ice lamellas can be realized on grooved surfaces with different groove widths. The preferential ice growth direction was applied to fabricate crisscross GO aerogel lamellar architecture with a negative Poisson ratio. The reviewer finds the issues listed below. Based on the issues, the reviewer suggests that the work be transferred to other journals with a more specific interest in material synthesis.

1. The novelty of the work is limited. In the literature, there are several works demonstrating the control of ice growth on grooved surfaces [refs. 19-23]. This current work can be considered as an extension of previous works in the literature focusing on, more specifically, studying the effect of the groove width and establishing a regime map. Therefore, compared to the existing works, the novelty of the work is limited.
2. The mechanism part is superficial. First, part of the content in the proposed freezing mechanism is simply the description of the observed phenomena. Second, the authors attributed the control of ice to the local temperature gradient on the surface. A similar mechanism has been proposed to explain the control of ice growth in the literature [24]; therefore, it is not novel. Third, the argued responsible mechanism is arbitrary because the authors did not provide reasons for ruling out other possible mechanisms, such as the variation of the nucleation energy barrier for grooves with different sizes for a nucleation-limited process [19].
3. The simulation results shown in Fig. 4 can not explain the obtained ice growth results. Fig. 4e shows that perpendicular ice growth is preferred for surfaces with a medium Λ value (with a large temperature gradient). However, perpendicular, parallel, and random ice growths were observed on the surfaces with a medium Λ , as shown in Figs. 2 and 3. Therefore, the simulation results in Fig. 4e can not explain the obtained experimental

results.

4. The conditions for conducting the experiments and simulations were not clear. What are the humidity, environmental temperature, and pressure? Is the surface temperature controlled during the experiments? What are the supersaturation values or the subcooling values in the experiments?

In the introduction:

5. "Lo. et al reported that.... preferentially grow along the direction perpendicular to V-shape grooves¹⁹. Munch. et al reported that ice crystals tend to orient parallel to the grooves during the freeze-casting process²³. Obviously, ice orientation on grooved surfaces is still under debate,.."

The two works both illustrated the parallel ice growth along the groove direction. Adding works showing the perpendicular ice growth would be more convincing for the argument that ice orientation on grooved surfaces is under debate.

In the section "Groove size effect...":

6. What are the sizes and shapes of the suspended graphene oxide (GO) nanosheets, alumina (Al₂O₃) platelets, and boron nitride (BN) nanosheets? Please give the details.

7. What are the dimensions of the grooves for the results shown in Fig. S3?

8. "Therefore, we infer that ice orientation is controlled by λ , $k_{\text{suspension}}$, and W/D ."

The argument that the three factors are responsible for ice control is arbitrary without sufficient evidence. Other factors that may affect ice growth were not discussed, for example, the variation of the nucleation energy barrier with different groove sizes.

9. The authors combine the effects of thermal conductivity and diagonal length into one factor, making it more challenging to understand the specific factor's influence on ice growth. For example, experiments should be controlled with a constant k value while varying λ values to know the effect of λ .

In the section "proposed freezing mechanism":

10. "For the grooved surface with small λ , ice crystals nucleate at the bottom of the grooves and grow randomly....For the grooved surface with large λ , ice crystals nucleate and grow randomly at the bottom of the grooves....."

The whole statement here is more like a phenomenon description, not a mechanism explanation. Besides, why does the ice crystal nucleate at the bottom of the groove but not at the top of the wall?

11. “ $(T_2-T_1)/(\lambda/2)$, representing the localized temperature gradient along the y axis (Fig. 4e). It shows that the perpendicular orientation (red line) can be achieved with grooves in a narrow range of λ , which can generate the temperature gradient large enough (>1.5 K/cm) to promote the preferential growth of ice crystals, representing another kind of groove size effect.”

Perpendicular, parallel, and random ice growths were observed on the surfaces with the given Lambda range, as shown in Figs. 2 and 3. Therefore, the simulation results in Fig. 4e can not explain the obtained experimental results.

12. The simulation results shown in Fig. 4 are the results after the ice formation on the surfaces. Using the temperature distribution after ice formation to infer the ice growth mechanism is inappropriate because the ice formation alters the surface temperature distribution. A figure showing the temperature profile before ice formation is recommended to better illustrate the effect of local temperature gradient on the icing process. This figure should be compared to the ice growth patterns in experiments. From the comparison, it would be more clear to understand the effect of the local temperature gradient on the icing process.

In the section “Unique porous material fabricated..”:

13. “As shown in Fig. 5h, our crisscross-aligned graphene aerogels exhibit an ultralow thermal conductivity (12.1 mW m⁻¹ K⁻¹) comparable to the commercial insulating aerogel.” The obtained thermal conductivity value is not particularly low, as listed in Table 2.

Reviewer #2 (Remarks to the Author):

The authors provide an experimental and theoretical investigation of the impact of grooved surfaces on the growth of ice crystals. Based on a review of the manuscript, the following revisions are provided:

1) The English grammar does need to be improved to bring the manuscript up to a publishable quality.

- 2) Given that the paper does generally discuss the freeze-casting process, the citations are relatively weak on the freeze casting process, and in particular on other techniques that align the structure of freeze cast materials. The authors should include additional citations on this topic and compare and contrast their materials from what is currently available in the literature.
- 3) When working with alignment of particles in suspensions via ice crystal growth, even when alignment of the crystals is shown (such as the results here where the crystals are aligned either perpendicular or parallel to the grooves) it is very rare for this alignment to be expressed throughout the entire structure due to the random growth of ice crystals. However, the authors present their results as though 100% of ice crystals are in alignment. The authors should provide some data and text that describes the proportion of ice crystals that are in alignment when they discuss the parallel or perpendicular orientations.
- 4) The authors present an interesting theory for the change in direction of ice growth, but a logical explanation would be that the change in orientation when changing the “W” parameter has to do with the relationship between the size of the ice crystals and the size of “W” (where, at certain small sizes, the ice crystals are unable to grow perpendicular to the grooves just due to geometric constraints. The authors should discuss this and either negate it in favor of their proposed mechanism or include it in the discussion.
- 5) Figure 3 is generally well constructed, but in the first two rows of diagrammed ice growth it’s difficult to see the ice crystals because their too small. The authors should increase their size to better convey their message.
- 6) The authors don’t discuss the replicates that were fabricated for this research. The authors must describe how many replicates were fabricated and performed (in terms of materials and test, respectively) to ensure that these results are reproducible and accurate.

Reviewer #3 (Remarks to the Author):

Ice formation on solid surface is ubiquitous both in nature and in our daily lives, which also plays a crucial role in various application. In this manuscript, the authors investigated ice orientation on grooved surface through in situ observation and discovered the size effect of surfaces grooves on ice orientation. In addition, they also demonstrated the application of their finding in material fabrication. I believe this discovery would provide powerful

guidance for the design of solid surface to control ice orientation for various engineering applications, as demonstrated in this work to fabricate aerogel with negative Poisson's ratio. I think this work exhibits obvious novelty in mechanism elaboration of fundamental physical process, material manufacturing and function exploration. Therefore, I would recommend its publication in Nature Communications. Some specific comments for the authors to further improve their work.

1. Why GO was selected as the probe for in situ observation? Whether the three growth modes of ice crystals can be observed when adding other materials as the probe?

2. What's the height of the surface grooves? Is it crucial? Whether other heights have been studied? Preferential ice orientation also occurs?

3. The material of commercial aerogel should be mentioned. Abbreviations like "GO-based aerogels" should be defined before use.

4. There are grammatical errors such as "It shows that the perpendicular orientation (red line) can be achieved with grooves in a narrow range of λ , which can generate the temperature gradient large enough (>1.5 K/cm) to promote the preferential growth of ice crystals, representing another kind of groove size effect". Overall, similar errors should be corrected in the whole manuscript.

Reviewer #4 (Remarks to the Author):

This work elaborated on how the surface groove sizes influence the orientation of ice crystals and further affect the pore structure and anisotropy of corresponding graphene aerogels. This discovery of the groove size effect is novel and exciting, especially the in-situ observation of the detailed freezing process by using fluorescence microscopy, which could eliminate the influence of freeze-drying on the pore structure of aerogel materials. This work could attract more attention for researchers dedicated to developing freeze-casted porous aerogels. However, the manuscript needs major revision, and some points must be addressed before published in Nature Communications, as follows:

1. Please add the temperature bar for the infrared images in Fig. 5h.

2. Here, the authors discussed a lot on the W and D. What about the influence of the thickness of the PDMS substrates? The PDMS is an insulation layer on the cold peltier and will somehow affect the cold source transfer. This should be mentioned in "Fabrication of

grooved PDMS substrate". The temperature of the cold source should be supplemented as well.

3. In Fig. 2, the author presented the materials with different thermal conductivities, showing different ice orientation results with W/D. However, in Fig. 1 and 5, different suspensions were used, one is GO suspension, and the other one is GO/PVA suspension. Please explain the difference in their thermal conductivities and why not keep the same with Fig. 1. The same case with the BN/PU suspension. Specifically, the freezing mechanism investigation and the final aerogels use different starting suspensions. Please explain the potential effects and if the proposed mechanism could be adopted in those practical scenarios.

4. Please add references for the thermal conductivities of those variable materials in Figure S2 if they are not the results measured in this work. Moreover, please specify whether they are the thermal conductivities of the raw 2D materials or related built materials (2D materials with polymers for better dispersion ability).

5. In Fig.2d, please explain the function of marked blue and red regions in the caption.

6. Please give more details of the commercial insulating aerogels used for comparison.

7. There are some sloppy grammatical issues and inappropriate expressions here and there. More work should be done on the modification of expression for better readability.

“Preferential ice growth on grooved surface for crisscross-aligned graphene aerogel with ultralow negative Poisson’s ratio”

by M. Li, N. Zhao, A. Mao, M. Wang, Z. Shao, W. Gao, and H. Bai

Response to Reviewers’ Comments

We would like to express our appreciation to the four reviewers for their very helpful comments on our paper. We realize that all of you spent a considerable amount of time carefully reading the manuscript and formulating a strategy of action for us to make this paper better. We are very grateful for this.

We have attempted to address all the issues raised in the review and to revise the paper exactly along the lines suggested. These revisions are marked in blue in the manuscript. Our responses to all the reviewers’ comments are provided below, beginning with a list of detailed revisions.

List of Detailed Revisions

1. In the authorship, Prof. Weiwei Gao has been added as co-corresponding author.
2. Page 1, the corresponding addresses 2 and 3 have been added in the revised manuscript.
3. Page 2, “Lo. et al reported that ... depending on the cold surface” and “we investigated ... a size effect of the grooves” have been modified in the revised manuscript.
4. Page 4, “As shown in Supplementary Figure 4 ... this curve exhibits multiple peaks” and “As shown in Supplementary Figure 5 ... with different $k_{\text{suspension}}$ ” have been added in the revised manuscript.
5. Page 5, “It should be noted that ... should be developed” has been added in the revised manuscript.
6. Page 5, Figure 2d has been modified in the revised manuscript.
7. Page 5, “According to the heterogeneous nucleation rate ... typical freezing processes on grooved surfaces with different sizes are illustrated in Fig. 3” has been added in the revised manuscript.
8. Page 6, “When $W \gg D$ or $W < D$, ice crystals nucleate and grow at the bottom of the grooves until they randomly merge together into a small-area alignment”, “which is a common growth mode for ice crystals on the grooved surface”, and “due to the geometric constraint by the grooves” have been added in the revised manuscript.
9. Page 6, Figure 3 has been modified.

10. Page 7, “explain the perpendicular ice growth”, “Since perpendicular orientation occurs on the grooved surfaces with $W \approx D$ ”, and “and $W \approx D$ ” have been added in the revised manuscript.
11. Page 7, “Additionally, in order to further ... generated to induce the perpendicular growth of ice crystals” has been added in the revised manuscript.
12. Page 9, “It is well-known that ... complex porous architectures” has been added in the revised manuscript.
13. Page 10, Figure 5 has been modified.
14. Page 11, “As shown in Fig. 5h, ... is broken after compression” has been modified in the revised manuscript.
15. Page 13, “The PDMS substrate is 1 mm in thickness and the grooves are 5 μm in height, respectively” has been added in the revised manuscript.
16. Page 14, “at a cooling rate of 5 $^{\circ}\text{C min}^{-1}$ from -30°C ”, “The commercial aerogel is silica aerogel fabricated by sol-gel method, having a thermal conductivity of around 16 $\text{mW m}^{-1} \text{K}^{-1}$ ”, and “All the experiments ... reliable data” have been added in the revised manuscript.
17. Page 16, references 24, 25, 26, and 27 have been added in the revised manuscript.
18. Page 17, “Fundamental Research Funds for the Central Universities (2021FZZX001-17) ” has been added in the revised manuscript.
19. Page 20, Supplementary Figure 2 has been modified.
20. Page 21, “on the grooved surfaces with $W = D = 10 \mu\text{m}$ ” has been added in the figure caption in the revised manuscript.
21. Supplementary Figures 4, 5, 7, and Table 1 have been added in the revised manuscript.

22. Page 33, Supplementary Table 3 has been revised in the manuscript.

Detailed Response to the Reviewers' Comments

Reviewer # 1:

Comment: *In this work, the authors investigated the ice growth on grooved surfaces. It was found that parallel and perpendicular ice lamellas can be realized on grooved surfaces with different groove widths. The preferential ice growth direction was applied to fabricate crisscross GO aerogel lamellar architecture with a negative Poisson ratio. The reviewer finds the issues listed below. Based on the issues, the reviewer suggests that the work be transferred to other journals with a more specific interest in material synthesis.*

Reply: We are very grateful for your insightful and constructive comments, which we find very helpful in revising our manuscript. However, we believe that in addition to create a novel approach for materials synthesis, we also propose a new mechanism of preferential ice growth, which will have a wide impact on various applications, such as freeze casting, anti-icing/deicing and organ cryo-preservation. Following your comments and suggestions, the freezing mechanism has been now more thoroughly studied with additional simulation and analysis.

We sincerely hope that we have addressed all your concerns in the revised manuscript.

Comment: *1. The novelty of the work is limited. In the literature, there are several works demonstrating the control of ice growth on grooved surfaces [refs. 19-23]. This current work can be considered as an extension of previous works in the literature focusing on, more specifically, studying the effect of the groove width and establishing a regime map. Therefore, compared to the existing works, the novelty of the work is limited.*

Reply: Thank you for your comment. Compared to previous studies, we think our work has sufficient novelty in the following aspects:

(1) For the first time, we accomplished the visualization of ice boundaries in water through adding a little amount of dye-labeled graphene oxide nanosheets as non-invasive probes, discovered and proved that surface grooves have a size effect on effectively controlling ice orientation.

(2) Additionally, a unique surface groove pattern was utilized to successfully fabricate a graphene aerogel with crisscross architecture and unprecedented negative Poisson's ratio.

(3) This work also highlights the possibility of utilizing the designability and versatility of groove patterning to program the preferential growth of ice crystals to fabricate materials with complex porous architectures and therefore unprecedented properties, which could attract more attention for researchers dedicated to developing freeze-casted porous aerogels.

We are also happy to see that all other reviewers have recognized the novelty of our work. As written by Reviewer 2: The authors present an interesting theory for the change in direction of ice growth; Reviewer 3: I think this work exhibits obvious novelty in mechanism elaboration of fundamental physical process, material manufacturing and function exploration; Reviewer 4: This discovery of the groove size effect is novel and exciting.

Comment: *2. The mechanism part is superficial. First, part of the content in the proposed freezing mechanism is simply the description of the observed phenomena. Second, the authors attributed the control of ice to the local temperature gradient on the surface. A similar mechanism has been proposed to explain the control of ice growth in the literature [24]; therefore, it is not novel. Third, the argued responsible mechanism is arbitrary because the authors did not provide reasons for ruling out other possible mechanisms, such as the variation of the nucleation energy barrier for grooves with different sizes for a nucleation-limited process [19].*

Reply: Thank you for your comment. We have added more discussions about the freezing mechanism in the revised manuscript. For example, other factors including

nucleation barrier and different alignment mechanism of ice crystals in related freeze casting methods have been further analyzed and discussed.

We note that temperature gradient is a well-known factor to control ice growth, as reported in reference [24] (Sci. Adv., 2015, 1, e1500849), where a macroscopic temperature gradient was established to achieve the preferential growth of ice crystals. However, we believe that our current work is novel because we report for the first time that a localized microscopic temperature gradient between two microgrooves can also induce the preferential growth of ice crystals.

Additionally, we consider that the grooves with different sizes in the current work have a negligible effect on their nucleation energy barrier. According to definition, the nucleation barrier (ΔG) is closely correlated to the size of surface microstructure, as $\Delta G = \Delta G_{homo} f(m, R')$, where $R' = R/R_c$, R_c is the critical size of ice nuclei, and R is the radius of the substrate (Chem. Soc. Rev. 2018, 47, 7116). It has been reported that the critical nucleus size is several nanometers (Nature 2019, 576, 437). The groove in our study is around several to tens of micrometer scale, which is three orders of magnitude higher than R_c . Therefore, as shown in the following figure adopted from the reference (Chem. Soc. Rev. 2018, 47, 7116), we consider that the groove size in our study (around 10 μm) has a negligible effect on the nucleation energy barrier.

We have added these discussions in the revised manuscript.

[FIGURE REDACTED]

Comment: 3. *The simulation results shown in Fig. 4 can not explain the obtained ice growth results. Fig. 4e shows that perpendicular ice growth is preferred for surfaces with a medium Lambda value (with a large temperature gradient). However, perpendicular, parallel, and random ice growths were observed on the surfaces with a medium Lambda, as shown in Figs. 2 and 3. Therefore, the simulation results in Fig. 4e can not explain the obtained experimental results.*

Reply: Thank you for your insightful comments. Additional simulation of the temperature field on the groove surfaces with a constant Lambda (50 μm) but

different W/D was performed. As shown in the following figure, for the grooves with $W = 25 \mu\text{m}$, $D = 5 \mu\text{m}$ and $W = 5 \mu\text{m}$, $D = 25 \mu\text{m}$, there is no significant temperature gradient to induce the preferential growth of ice crystals. For the grooves with $W = D = 15 \mu\text{m}$, an obvious temperature gradient was generated to induce the perpendicular growth of ice crystals.

We have added these discussions and the following figure as Figure S7 in the revised manuscript.

Comment: 4. The conditions for conducting the experiments and simulations were not clear. What are the humidity, environmental temperature, and pressure? Is the surface temperature controlled during the experiments? What are the supersaturation values or the subcooling values in the experiments?

Reply: Thank you for your comment. The relative humidity is around 60%, the environment temperature is around 25°C , and the pressure is 1 atm. Surface temperature was decreased at a speed of $5^\circ\text{C}/\text{min}$ from -30°C . These conditions are the same for the experiments and simulations.

We have provided this information in the revised manuscript.

Comment: 5. In the introduction: “Lo. et al reported that.... preferentially grow along the direction perpendicular to V-shape grooves¹⁹. Munch. et al reported that ice crystals tend to orient parallel to the grooves during the freeze-casting process²³. Obviously, ice orientation on grooved surfaces is still under debate,..”

The two works both illustrated the parallel ice growth along the groove direction. Adding works showing the perpendicular ice growth would be more convincing for the argument that ice orientation on grooved surfaces is under debate.

Reply: Thank you for your comment. In reference 19 (ACS Nano, 2017, 11, 2665), ice basal facets aligned perpendicularly to the groove direction and ice crystals aligned parallelly to the groove direction at the macroscopic level, as shown in the following figure.

[FIGURE REDACTED]

In reference 23 (J. Am. Ceram. Soc., 2009, 92, 1534), ice crystals align randomly on a smooth surface (figure a), and ice crystals align perpendicular (figure b) or parallel to (figure c) the surface grooves.

Therefore, ice orientation on grooved surfaces is still under debate.

[FIGURE REDACTED]

Comment: 6. In the section “Groove size effect...”:What are the sizes and shapes of the suspended graphene oxide (GO) nanosheets, alumina (Al₂O₃) platelets, and boron nitride (BN) nanosheets? Please give the details.

Reply: Thank you for your comment. The detailed information of the suspended nanosheets/platelets are listed in the following table. This has been added as Table S1 in the revised manuscript.

Medium	Shape	Thickness (nm)
GO	Nanosheet	0.8 - 1.2
Alumina	Platelet	~250
BN	Nanosheet	~30

Comment: 7. In the section “Groove size effect...”: What are the dimensions of the grooves for the results shown in Fig. S3?

Reply: Thank you for your comment. The width and distance of the grooves for the results shown in Fig. S3 are both 10 μm .

We have added this information in the figure caption of Fig. S3 in the revised manuscript.

Comment: 8. In the section “Groove size effect...”: “Therefore, we infer that ice orientation is controlled by λ , $k_{\text{suspension}}$, and W/D .”

The argument that the three factors are responsible for ice control is arbitrary without sufficient evidence. Other factors that may affect ice growth were not discussed, for example, the variation of the nucleation energy barrier with different groove sizes.

Reply: Thank you for your comment. In our manuscript, λ , $k_{\text{suspension}}$, and W/D are three critical factors controlling ice orientation. As for ice nucleation, we have analyzed in Comment 2 that the effect of the variation of ice nucleation barrier on the substrate with different groove sizes can be neglected.

Comment: 9. In the section “Groove size effect...”: *The authors combine the effects of thermal conductivity and diagonal length into one factor, making it more challenging to understand the specific factor’s influence on ice growth. For example, experiments should be controlled with a constant k value while varying λ values to know the effect of λ .*

Reply: Thank you for your comment. The relationship between ice orientation with a constant k value and various λ is shown in the following detailed figure. The effect of λ on ice orientation obeys the similar rules for the suspension with different k . For large and small λ , ice crystals tend to grow into small-area alignment. For medium λ , the alignment of ice crystals depends on the ratio of W and D .

We have added the following figure as Figure S5 in the revised manuscript.

Comment: 10. In the section “proposed freezing mechanism”: “For the grooved surface with small λ , ice crystals nucleate at the bottom of the grooves and grow randomly....For the grooved surface with large λ , ice crystals nucleate and grow randomly at the bottom of the grooves.....”

The whole statement here is more like a phenomenon description, not a mechanism explanation. Besides, why does the ice crystal nucleate at the bottom of the groove but not at the top of the wall?

Reply: Thank you for your comment. We have added more discussions of the freezing mechanism in the revised manuscript. Additionally, ice crystals nucleate at the bottom of the groove where a lower temperature is reached due to both the groove height and the poor thermal conductivity of PDMS substrate. We have added these discussions in the revised manuscript.

Comment: 11. In the section “proposed freezing mechanism”: “ $(T_2-T_1)/(\lambda/2)$, representing the localized temperature gradient along the y axis (Fig. 4e). It shows that the perpendicular orientation (red line) can be achieved with grooves in a narrow range of λ , which can generate the temperature gradient large enough (>1.5 K/cm) to promote the preferential growth of ice crystals, representing another kind of groove size effect.”

Perpendicular, parallel, and random ice growths were observed on the surfaces with the given Lambda range, as shown in Figs. 2 and 3. Therefore, the simulation results in Fig. 4e can not explain the obtained experimental results.

Reply: Thank you for your comments. As shown in Comment 3, we have added additional simulation and analysis to explain the obtained experimental results. For the grooves with $W = 25 \mu\text{m}$, $D = 5 \mu\text{m}$ and $W = 5 \mu\text{m}$, $D = 25 \mu\text{m}$, there is no significant temperature gradient to induce the preferential growth of ice crystals. For the grooves with $W = D = 15 \mu\text{m}$, an obvious temperature gradient was generated to induce the perpendicular growth of ice crystals. We have added these discussions in the revised manuscript.

Comment: 12. In the section “proposed freezing mechanism”: The simulation results shown in Fig. 4 are the results after the ice formation on the surfaces. Using the temperature distribution after ice formation to infer the ice growth mechanism is inappropriate because the ice formation alters the surface temperature distribution. A figure showing the temperature profile before ice formation is recommended to better illustrate the effect of local temperature gradient on the icing process. This figure should be compared to the ice growth patterns in experiments. From the comparison, it would be more clear to understand the effect of the local temperature gradient on the icing process.

Reply: Thank you for your comment. We agree that ice formation alters the surface temperature distribution. Actually, we did already include phase change module in our simulation process, in order to consider the temperature distribution after ice formation. Upon your request, a figure showing the temperature profile before ice formation in the GO suspension is provided below, where the localized temperature gradient is the same as that after ice formation.

Comment: 13. In the section “Unique porous material fabricated..”: “As shown in Fig. 5h, our crisscross-aligned graphene aerogels exhibit an ultralow thermal conductivity (12.1 mW m⁻¹ K⁻¹) comparable to the commercial insulating aerogel.”

The obtained thermal conductivity value is not particularly low, as listed in Table 2.

Reply: Thank you for your comment. The thermal conductivity of 4.78 mW m⁻¹ K⁻¹ in references 39 was measured under vacuum condition. Therefore, our crisscross-aligned graphene aerogel exhibits excellent thermal insulation property compared to similar aerogels reported in the literature.

We have labelled the condition for the measurement of thermal conductivity of the aerogels in the revised manuscript.

Reviewer # 2:

Comment: *The authors provide an experimental and theoretical investigation of the impact of grooved surfaces on the growth of ice crystals. Based on a review of the manuscript, the following revisions are provided:*

Reply: We are very grateful for your insightful and constructive comments, which we find very helpful in revising our manuscript.

Comment: *1. The English grammar does need to be improved to bring the manuscript up to a publishable quality.*

Reply: Thank you for your suggestion. We have tried our best to improve the English grammar in our revised manuscript.

Comment: *2. Given that the paper does generally discuss the freeze-casting process, the citations are relatively weak on the freeze casting process, and in particular on other techniques that align the structure of freeze cast materials. The authors should include additional citations on this topic and compare and contrast their materials from what is currently available in the literature.*

Reply: Thank you for your helpful suggestion. Accordingly, we have added more references and discussions to further compare the mechanism and potential of our method with other important studies of freeze casting. Specifically, external (magnetic) field assisted freeze casting, underflow freezing, bidirectional freezing, surface wettability induced freezing are discussed with their advantages and limitations. Briefly, we believe that the groove size effect that we report here is novel and inspiring for the construction of complex porous architectures. Moreover, on the material property side, a crisscross-aligned graphene aerogel with an ultralow negative Poisson's ratio was fabricated, which was difficult to realize by previous methods.

We have added these discussions in the revised manuscript.

Comment: 3. When working with alignment of particles in suspensions via ice crystal growth, even when alignment of the crystals is shown (such as the results here where the crystals are aligned either perpendicular or parallel to the grooves) it is very rare for this alignment to be expressed throughout the entire structure due to the random growth of ice crystals. However, the authors present their results as though 100% of ice crystals are in alignment. The authors should provide some data and text that describes the proportion of ice crystals that are in alignment when they discuss the parallel or perpendicular orientations.

Reply: Thank you for raising this very important issue. We agree it is unlikely that 100% of ice crystals are in alignment. In order to investigate the proportion of ice crystals that are in alignment, SEM images of the resulted porous architecture were processed with Image J in the following procedure. As shown below, when ice crystals are in alignment, there are two main peaks in the gray value-angle curve, indicating that gray regions distribute along a single direction; when ice crystals are randomly aligned, there are multiple peaks in the curve, indicating that gray regions distribute randomly.

We have added these discussions and the following figure as Figure S4 in the revised manuscript.

Comment: 4. *The authors present an interesting theory for the change in direction of ice growth, but a logical explanation would be that the change in orientation when changing the “W” parameter has to do with the relationship between the size of the ice crystals and the size of “W” (where, at certain small sizes, the ice crystals are unable to grow perpendicular to the grooves just due to geometric constraints. The authors should discuss this and either negate it in favor of their proposed mechanism or include it in the discussion.*

Reply: Thank you for raising this very important issue. In our manuscript, W refers to the width of convex part of the groove, and D refers to the width of the concave part. We agree that when D is small, ice crystals grow parallel to the grooves due to the geometric constraints. When D is large, changing D would change the temperature distribution and further make ice crystals grow perpendicular to the grooves under localized temperature gradient. Therefore, there are two mechanisms to control the parallel and perpendicular growth of ice crystals.

We have added these discussions in the revised manuscript.

Comment: 5. *Figure 3 is generally well constructed, but in the first two rows of diagrammed ice growth it’s difficult to see the ice crystals because their too small. The authors should increase their size to better convey their message.*

Reply: Thank you for your suggestion. We have modified this figure in the revised manuscript for better visualization of the ice crystals.

Comment: 6. The authors don't discuss the replicates that were fabricated for this research. The authors must describe how many replicates were fabricated and performed (in terms of materials and test, respectively) to ensure that these results are reproducible and accurate.

Reply: Thank you for your suggestions. For each category, at least five replicates were prepared to obtain reliable statistical data. The orientation of ice crystals was summarized by fabricating at least five samples on each grooved surface with various parameters to ensure the reproducibility and accurate of these results. We have added these descriptions in the revised manuscript.

Reviewer # 3:

Comment: *Ice formation on solid surface is ubiquitous both in nature and in our daily lives, which also plays a crucial role in various application. In this manuscript, the authors investigated ice orientation on grooved surface through in situ observation and discovered the size effect of surfaces grooves on ice orientation. In addition, they also demonstrated the application of their finding in material fabrication. I believe this discovery would provide powerful guidance for the design of solid surface to control ice orientation for various engineering applications, as demonstrated in this work to fabricate aerogel with negative Poisson's ratio. I think this work exhibits obvious novelty in mechanism elaboration of fundamental physical process, material manufacturing and function exploration. Therefore, I would recommend its publication in Nature Communications. Some specific comments for the authors to further improve their work.*

Reply: Thank you for your positive rating of our work. Following your constructive comments and suggestions, we have revised our manuscript and further improved our work.

Comment: *1. Why GO was selected as the probe for in situ observation? Whether the three growth modes of ice crystals can be observed when adding other materials as the probe?*

Reply: Thank you for your comment. GO nanosheet has a single atom layer, therefore when adding the same number of probes for observation, GO nanosheets have a lower weight, which wouldn't affect the ice growth. We also added small amount of dye-labelled polystyrene particles into the suspension as the probe of ice boundary. As shown in the following figure, three growth modes of ice crystals can be also observed.

Comment: 2. *What's the height of the surface grooves? Is it crucial? Whether other heights have been studied? Preferential ice orientation also occurs?*

Reply: Thank you for your comment. The height of the surface grooves is 5 μm . We also studied the grooves with 25 μm and 50 μm in height, which can also induce the preferential orientation of ice crystals. We have added the information in the revised manuscript.

Comment: 3. *The material of commercial aerogel should be mentioned. Abbreviations like "GO-based aerogels" should be defined before use.*

Reply: Thank you for your comment. The commercial aerogel is composed of silica. Accordingly, we have defined all abbreviations in the revised manuscript.

Comment: 4. *There are grammatical errors such as "It shows that the perpendicular orientation (red line) can be achieved with grooves in a narrow range of λ , which can generate the temperature gradient large enough ($>1.5 \text{ K/cm}$) to promote the*

preferential growth of ice crystals, representing another kind of groove size effect”.

Overall, similar errors should be corrected in the whole manuscript.

Reply: Thank you for your comment. We have corrected the grammatical errors in the revised manuscript.

Reviewer # 4:

Comment: *This work elaborated on how the surface groove sizes influence the orientation of ice crystals and further affect the pore structure and anisotropy of corresponding graphene aerogels. This discovery of the groove size effect is novel and exciting, especially the in-situ observation of the detailed freezing process by using fluorescence microscopy, which could eliminate the influence of freeze-drying on the pore structure of aerogel materials. This work could attract more attention for researchers dedicated to developing freeze-casted porous aerogels. However, the manuscript needs major revision, and some points must be addressed before published in Nature Communications, as follows:*

Reply: Thank you very much for recognizing the novelty of both our research method and discovery. We have revised our manuscript following your comments and suggestions.

Comment: *1. Please add the temperature bar for the infrared images in Fig. 5h.*

Reply: Thank you for your comment. The temperature bar for the infrared images in Fig. 5h has been added in the revised manuscript.

Comment: *2. Here, the authors discussed a lot on the W and D. What about the influence of the thickness of the PDMS substrates? The PDMS is an insulation layer on the cold peltier and will somehow affect the cold source transfer. This should be mentioned in "Fabrication of grooved PDMS substrate". The temperature of the cold source should be supplemented as well.*

Reply: Thank you for your comment. We agree that the PDMS is an insulation layer and affects the cold source transfer. Therefore, the thickness should be as small as possible and 1 mm was selected here. The temperature of the cold source was set to decrease with a speed of 5 °C/min from -30°C. These details have been added in the revised manuscript.

Comment: 3. *In Fig. 2, the author presented the materials with different thermal conductivities, showing different ice orientation results with W/D. However, in Fig. 1 and 5, different suspensions were used, one is GO suspension, and the other one is GO/PVA suspension. Please explain the difference in their thermal conductivities and why not keep the same with Fig. 1. The same case with the BN/PU suspension. Specifically, the freezing mechanism investigation and the final aerogels use different starting suspensions. Please explain the potential effects and if the proposed mechanism could be adopted in those practical scenarios.*

Reply: Thank you for raising this very important issue. In Fig. 1, the content of GO used to label ice boundaries should be as low as possible to avoid affecting ice growth.

Based on the in situ observation, we proposed that surface grooves at the microscopic level have a size effect on efficiently controlling ice growth, which is closely related to the thermal conductivity of the freezing suspension. With higher thermal conductivity, smaller λ is required to effectively generate preferential orientation. For example, if the thermal conductivity of our suspension is higher than $60 \text{ W m}^{-1} \text{ K}^{-1}$, nanosized grooves are required according to our theory. However, other mechanisms like nucleation barrier difference may play a more important role with nanosized grooves and new mechanism should be developed. Therefore, we anticipate that our theory is valid only when the thermal conductivity of suspension is not too high and only micrometer sized grooves are required. Fortunately, for common occasions including the suspensions like GO/PVA and BN/PU that we used, their thermal conductivities are below $10 \text{ W m}^{-1} \text{ K}^{-1}$, where our theory is applicable.

We have added this discussion in the revised manuscript.

Comment: 4. *Please add references for the thermal conductivities of those variable materials in Figure S2 if they are not the results measured in this work. Moreover, please specify whether they are the thermal conductivities of the raw 2D materials or related built materials (2D materials with polymers for better dispersion ability).*

Reply: Thank you for your comment. The values shown in Figure S2 are actually the thermal conductivities of the suspensions used in our study. We have added the information in the revised manuscript to avoid misunderstanding.

Comment: 5. In Fig.2d, please explain the function of marked blue and red regions in the caption.

Reply: Thank you for your comment. We intended to highlight the perpendicular ice orientation with marked blue regions and parallel ice orientation with marked red regions, respectively. In order to avoid misunderstanding, Fig. 2d has been revised as following.

Comment: 6. *Please give more details of the commercial insulating aerogels used for comparison.*

Reply: Thank you for your comment. The commercial insulating aerogels are fabricated with silica through sol-gel method, having a thermal conductivity of around $16 \text{ mW m}^{-1} \text{ K}^{-1}$. We have added the information in the revised manuscript.

Comment: 7. *There are some sloppy grammatical issues and inappropriate expressions here and there. More work should be done on the modification of expression for better readability.*

Reply: Thank you for your suggestions. We have tried our best to revise the grammatical issues and expressions. Hope you will find the revised manuscript more readable.

REVIEWERS' COMMENTS

Reviewer #1 (Remarks to the Author):

The reviewer appreciates the authors' effort in revising their manuscript. Some of my previously raised concerns have been addressed in the revised version. However, the reviewer still considers the novelty of the work limited. This is based on (1) the control of ice orientation on microgroove-patterned surfaces has been demonstrated [refs. 19-23]., and (2) the proposed temperature gradient mechanism has also been shown in the literature [ref. 24]. The work further studies the effect of groove width on ice orientation and is considered an extension of previous works. Therefore, compared to the existing works, the novelty of the work is limited.

Reviewer #2 (Remarks to the Author):

The authors have addressed my concerns and I would suggest the manuscript be accepted.

Reviewer #3 (Remarks to the Author):

The authors have addressed all the comments. Now it can be accepted.

Reviewer #4 (Remarks to the Author):

The authors have addressed my suggestions, and the manuscript is ready for publication.

“Preferential ice growth on grooved surface for crisscross-aligned graphene aerogel with large negative Poisson’s ratio”

by M. Li, N. Zhao, A. Mao, M. Wang, Z. Shao, W. Gao, and H. Bai

Reviewer # 1:

Comment: *The reviewer appreciates the authors’ effort in revising their manuscript. Some of my previously raised concerns have been addressed in the revised version. However, the reviewer still considers the novelty of the work limited. This is based on (1) the control of ice orientation on microgroove-patterned surfaces has been demonstrated [refs. 19-23]., and (2) the proposed temperature gradient mechanism has also been shown in the literature [ref. 24]. The work further studies the effect of groove width on ice orientation and is considered an extension of previous works. Therefore, compared to the existing works, the novelty of the work is limited.*

Reply: Thank you for your comment. As we have mentioned in the introduction part, ice formation on the grooved surface is a ubiquitous phenomenon and ice orientation has been the focus of research in various aspects, including anti-icing/deicing, cell preservation, and fabrication of multifunctional materials, which are related to ice adhesion, cell survival, and formation of porous structure, respectively. Therefore, the efficient control of ice orientation is necessary but hardly realized due to the limited exploration of the mechanism of ice formation. In this work, we report for the first time that a localized microscopic temperature gradient between two microgrooves can induce the preferential growth of ice crystals and discover the size effect of surface grooves to guide the efficient control of ice orientation, which is a remarkable breakthrough in ice-related research field. Therefore, we think our work overcome the long-time existed bottleneck of ice formation in previous works and our contribution and discovery is novel.